# Feasibility-Aware Decision-Focused Learning for Predicting Parameters in the Constraints

**Jayanta Mandi**
Department of Computer Science
KU Leuven, Leuven, Belgium
jayanta.mandi@kuleuven.be

**Marianne Defresne**[*]
LAAS-CNRS, Université de Toulouse
CNRS, INSA, Toulouse
marianne.defresne@laas.fr

**Senne Berden**
Department of Computer Science
KU Leuven, Leuven, Belgium
senne.berden@kuleuven.be

**Tias Guns**
Department of Computer Science
KU Leuven, Leuven, Belgium
tias.guns@kuleuven.be

## Abstract

When some parameters of a constrained optimization problem (COP) are uncertain, this gives rise to a predict-then-optimize (PtO) problem, comprising two stages: the *prediction* of the unknown parameters from contextual information and the subsequent *optimization* using those predicted parameters. Decision-focused learning (DFL) implements the first stage by training a machine learning (ML) model to optimize the quality of the decisions made using the predicted parameters. When the predicted parameters occur in the constraints, they can lead to infeasible solutions. Therefore, it is important to simultaneously manage both feasibility and decision quality. We develop a DFL framework for predicting constraint parameters in a generic COP. While prior works typically assume that the underlying optimization problem is a linear program (LP) or integer LP (ILP), our approach makes no such assumption. We derive two novel loss functions based on maximum likelihood estimation (MLE): the first one penalizes infeasibility (by penalizing predicted parameters that lead to infeasible solutions), while the second one penalizes suboptimal decisions (by penalizing predicted parameters that make the true optimal solution infeasible). We introduce a single tunable parameter to form a weighted average of the two losses, allowing decision-makers to balance suboptimality and feasibility. We experimentally demonstrate that adjusting this parameter provides decision-makers control over this trade-off. Moreover, across several COP instances, we show that adjusting the tunable parameter allows a decision-maker to prioritize either suboptimality or feasibility, outperforming the performance of existing baselines in either objective.

## 1 Introduction

Many real-world optimization problems involve parameters that are unknown at decision time, such as uncertain customer demand in manufacturing problems or unknown traffic in delivery routing. This gives rise to *predict-then-optimize* (PtO) problems [22], where a machine learning (ML) model first makes point predictions of the unknown parameters from contextual features, after which the resulting instantiated optimization problem is solved.

---

[*]Affiliated to KU Leuven during submission of this article

39th Conference on Neural Information Processing Systems (NeurIPS 2025).

In *prediction-focused learning* (PFL), the predictive model is trained solely to maximize the accuracy of parameter predictions by minimizing standard ML losses. By doing so, PFL does not consider the effect of prediction errors on the solution to the optimization problem. For example, consider a vehicle routing problem, where one first needs to predict unknown customer demands, to then make routing decisions. In this setting, overestimations and underestimations can have differing effects. Overestimating demand may lead to a conservative suboptimal decision, while underestimating demand may lead to an infeasible decision. However, these differing effects are not accounted for by PFL losses.

To address this shortcoming of PFL, *decision-focused learning* (DFL) directly trains the ML model to minimize a *task-specific decision loss* that reflects the quality of the solution produced by using the predicted parameters. With a slight abuse of notation, we will use the term 'predicted solution' to denote the solution produced by using the predicted parameters, although the ML model does not *directly* predict the solution, it predicts parameters that are passed to an optimization solver at inference time.

Most existing DFL works [12, 13, 23, 33] focus on cases where the predicted parameters appear *only* in the objective function. In such settings, the decision loss is typically the *regret*, which measures suboptimality of the predicted solution. Instead, we focus on predicting parameters that appear in the constraints of the optimization problem. In this setting too, suboptimal solutions may arise, because the feasible space defined by the predicted parameters may exclude the true optimal solution. However, focusing only on suboptimality is insufficient. This is because the predicted solution may become infeasible with respect to the constraints instantiated by the *true* parameters. Therefore, the ML model should also be trained to minimize the likelihood of such infeasible outcomes.

Focusing only on suboptimality or on infeasibility generally does not lead to a desirable balance between these two objectives. To address this, Hu et al. [14] model decision-making with uncertain constraint parameters as a two-stage process: first, a solution is generated using the predicted parameters; then, if needed, corrective actions – incurring additional costs – are carried out to resolve potential infeasibility. Subsequent works [15, 16] have extended this line of research.

One limitation of their approach is that the corrective action is often problem-specific, and restricted to a specific form of optimization problem (e.g., a linear program). Instead, we avoid the complexity of a second stage altogether, and focus on predicting parameters that minimize both the infeasibility and the suboptimality of the first-stage solution itself. To this end, we first propose two novel loss functions: the Infeasibility Penalty Loss (IPL) and Optimality-Preserving Loss (OPL). The idea behind IPL is that when solving with the predicted parameters leads to a solution that is infeasible according to the true parameters, such infeasibility should be (proportionally) penalized. On the other hand, the OPL loss will penalize predicted parameters that make the true optimal solution infeasible, to avoid obtaining a suboptimal solution when solving with the predicted parameters.

These two losses often contradict each other. The IPL term encourages conservative parameter predictions to guarantee feasibility, even at the expense of producing extremely suboptimal solutions. In contrast, OPL incentivizes loose parameter predictions to ensure the optimal solution is not cut away. Moreover, in practice, a decision-maker may favor more feasibility over optimality or vice versa. To support this subjective preference of one loss over the other, we propose to minimize a weighted average of the two losses. In this way, we introduce a tunable weighting mechanism that adjusts the total loss based on the decision-maker's preference. We refer to our approach as *Odece*, for "optimizing decisions through end-to-end constraint estimation".

We compare Odece with the PFL approach of minimizing the mean squared error (MSE), and with some existing DFL approaches. These approaches do not allow the decision-maker to adjust the predicted solution based on their subjective preference between optimality and feasibility. Moreover, the DFL approaches for constraint prediction are limited to linear programs (LPs) or integer or mixed-integer LPs (ILPs and MILPs), or they focus on predicting full constraint systems rather than individual parameters within the constraints. Our experimental results across various optimization problem instances show that, for a single value of the tunable parameter, Odece matches or outperforms existing baselines in terms of both suboptimality and feasibility. More importantly, they show that our approach offers a simple way for decision-makers to *control* the trade-off between suboptimality and infeasibility using a single parameter.

## 2 Related Literature

**Differentiating through optimization problems.** Differentiating through optimization problems has gained attention in ML, as it allows solvers to be embedded into gradient-based training. This is often done by ADMM [9, 29] or by implicit differentiation [8] of the optimality conditions. Implicit differentiation techniques have been developed for quadratic programs [2] and conic optimization [1]. However, for combinatorial optimization problems like ILPs, the true gradient is zero almost everywhere because small changes in the parameters usually do not affect the optimal solution. Hence, implicit differentiation techniques, which are applicable to smooth optimization problems, cannot be applied to such combinatorial problems. To address this, Paulus et al. [26] propose CombOptNet, a technique for ILPs to compute pseudo-gradients based on the Euclidean distance between the solution and the constraints.

**Predicting objective parameters.** Most DFL works focus on predicting parameters in the objective function. Some of these works are based on differentiating through the problem after smoothing it, either using regularization [13, 18, 33] or perturbation [5, 10, 25, 28]. Another approach is to define informative surrogates of the decision loss [4, 12, 19, 20, 23, 31]. Regularization-based methods require differentiable solvers, such as *Cvxpylayers* [1] or *OptNet* [2]. In contrast, perturbation and surrogate-based approaches only need access to solutions. These solutions generally do not need to come from an exact solver and can be obtained by heuristics or alternative methods. For instance, surrogate-based techniques have been applied to planning problems solved via approximate automated planners [21]. We refer readers to a recent survey paper [22] for a more comprehensive overview.

**Predicting constraint parameters.** Relatively few DFL papers have considered the prediction of parameters in the constraints. Most of these [14–16] model decision-making as a two-stage process. In the first stage, a solution is obtained using the predicted parameters, which may be infeasible with respect to the true parameters. The second stage then applies a corrective action to convert the first-stage solution into a feasible one. Among these works, Hu et al. [15] also recommend a strategy for applying their approach to settings where no corrective action is available at inference time. Their proposal is to assign a very high penalty factor to the corrective action during training. In contrast to this line of work, we do not introduce corrective actions even during training and focus on balancing decision quality and feasibility in a single stage. Another approach [11] simultaneously learns the objective and the constraints, but the constraints are soft and the problem must be formulated as a discrete graphical model. The work by Nandwani et al. [24] aims to learn the full set of constraints from the features. They do this by learning a set of hyperplanes that separate the optimal solution from other *negative assignments*. A key component of their technique is the generation of such negative assignments, which must include both infeasible as well as feasible but suboptimal ones. In contrast to their setting, we focus on learning only the parameters within a known constraint structure. Their method does not easily generalize to our setting, in which the learned parameters must remain compatible with the known constraints to make high-quality solutions. Moreover, both Hu et al. [15] and Nandwani et al. [24] restrict their methods to LPs and ILPs, while the framework proposed in this paper is more general and applicable to a broader class of optimization problems. The prediction of constraints is also central in constraint acquisition, for optimization [3, 17] or constraint satisfaction [6, 32] problems, but the setting differs because the structure of the constraints is unknown and the uncertainty does not depend on any input features.

## 3 Preliminaries

**Problem setup.** We will consider a constrained optimization problem (COP) of the following form:

$$\min_{\boldsymbol{x} \in \Omega} \quad f(\boldsymbol{x}; \boldsymbol{q}) \tag{1a}$$

$$\text{subject to} \quad g_i(\boldsymbol{x}; \boldsymbol{\rho}) \leq 0, \qquad i \in \{1, \ldots, M\} \tag{1b}$$

where $\Omega$ defines the domain of the decision variable vector $\boldsymbol{x}$. $\Omega$ may be a subset of real numbers or integers, depending on the type of optimization problem. The upper and lower bounds for each decision variable are also specified by $\Omega$. We will assume that the constraint functions $g_i(\boldsymbol{x}; \boldsymbol{\rho})$ are differentiable functions. We do not explicitly express equality constraints, since any constraint

of the form $h_j(\boldsymbol{x}; \boldsymbol{\rho}) = 0$ can be represented by two inequality constraints: $h_j(\boldsymbol{x}; \boldsymbol{\rho}) \leq 0$ and $-h_j(\boldsymbol{x}; \boldsymbol{\rho}) \leq 0$. To make the notation simpler, we denote the set of constraints by $\mathcal{S}$ and the feasible set when the constraint parameters take the value $\rho$ by $\mathcal{F}(\rho)$. More formally,

$$\mathcal{F}(\boldsymbol{\rho}) = \{\boldsymbol{x} \in \Omega : g_i(\boldsymbol{x}; \boldsymbol{\rho}) \leq 0, \quad \forall i \in \mathcal{S}\} \tag{2}$$

We use $\boldsymbol{x}^\star(\boldsymbol{q}, \boldsymbol{\rho})$ to denote a (possibly non-unique) optimal solution to the optimization problem for a given objective parameter $\boldsymbol{q}$ and constraint parameter $\boldsymbol{\rho}$. When multiple optimal solutions exist, we define the set of all optimal solutions as $\boldsymbol{W}(\boldsymbol{q}, \boldsymbol{\rho}) = \{\boldsymbol{x} : f(\boldsymbol{x}; \boldsymbol{q}) \leq f(\boldsymbol{x}'; \boldsymbol{q}); \ \forall \boldsymbol{x}' \in \mathcal{F}(\boldsymbol{\rho})\}$. Note that $\boldsymbol{W}(\boldsymbol{q}, \boldsymbol{\rho})$ is a subset of the feasible region $\mathcal{F}(\boldsymbol{\rho})$. So, if $\mathcal{F}$ is empty (i.e., no feasible solution exists), $\boldsymbol{W}$ is also empty. In the special case of a unique optimal solution, $\boldsymbol{W}(\boldsymbol{q}, \boldsymbol{\rho})$ is a singleton. However, our implementation relies on a solver that returns a single solution. When multiple optimal solutions exist, the commercial solvers we use break ties using a predefined rule.

**Example.** Consider the multi-dimensional 0-1 knapsack problem (MDKP) [27]. The objective of this problem is to choose a subset with maximal value from a given set of items, subject to $M$ capacity constraints. Let $\rho_i$ be the capacity in dimension $i$. There are $N$ items, the value of each item is $q_n$, and $\rho_{(n,i)}$ is the weight of item $n$ in dimension $i$. Each decision variable $x_n$ can be either zero or one, hence $\Omega = \{0, 1\}^N$. This optimization problem can be modeled as an ILP as follows:

$$\min_{x_{1:N}} \sum_{n=1}^{N} (-q_n) x_n \quad \text{such that} \quad x_n \in \{0, 1\} \ \forall n \in [N], \quad \sum_{n=1}^{N} \rho_{ni} x_n \leq \rho_i \ \forall i \in [M] \tag{3}$$

Here, $\mathcal{S}$ consists of the $M$ capacity constraints.

**Predict-then-Optimize problems.** In the PtO formulation for uncertainty in the constraints, some or all components of the parameter vector $\boldsymbol{\rho}$ are unknown and must be estimated before solving the COP. For example, in the knapsack problem, the item weights $\rho_{(n,i)}$ may be unknown, but the capacity in each dimension $\rho_i$ might be known. Additionally, we observe a set of values correlated with $\boldsymbol{\rho}$, which we refer to as *features* and denote by $\phi$. These features are used to make estimates of the unknown parameters. For the estimation task, a predictive model $\mathcal{M}_\theta$, parameterized by $\theta$, is trained using training data of the form $\mathcal{D} \equiv \{(\phi_\kappa, \boldsymbol{\rho}_\kappa, \boldsymbol{q}_\kappa, \boldsymbol{x}^\star(\boldsymbol{q}_\kappa, \boldsymbol{\rho}_\kappa))\}_{\kappa=1}^{K}$. Formally, the goal is to learn a mapping from the feature space to the parameter space. In this work, we assume that the cost vector $\boldsymbol{q}_\kappa$ is instance-specific but known and not used to learn the mapping. Note that the structure of the COP is fixed, i.e., the number of constraints and the domains of decision variables are known. Once trained, given a new feature vector $\phi$ and a known cost vector $\boldsymbol{q}$, the model produces an estimated parameter vector $\hat{\boldsymbol{\rho}} = \mathcal{M}_\theta(\phi)$ using $\phi$. Since this is a PtO task, $\mathcal{M}_\theta$ is ultimately evaluated based on the quality of the solution $\boldsymbol{x}^\star(\boldsymbol{q}, \hat{\boldsymbol{\rho}})$ induced by the predicted parameters.

**A maximum likelihood perspective on DFL.** We will develop a framework based on maximum likelihood estimation (MLE) to predict constraint parameters. Our goal is to maximize the log-likelihood of the observed data $\mathcal{D}$. In PFL, the parameters $\theta$ of the predictive model would be estimated to increase the likelihood of the observations $\boldsymbol{\rho}_\kappa$ given the features $\phi_\kappa$. However, in DFL training, the focus is on the likelihood of $\boldsymbol{x}^\star(\boldsymbol{q}_\kappa, \boldsymbol{\rho}_\kappa)$ being the optimal solution, given $\boldsymbol{q}_\kappa$ and $\phi_\kappa$. To put it another way, the MLE perspective of DFL consists of learning a $\theta$ to predict the constraint parameter vector $\hat{\boldsymbol{\rho}}_\kappa$ so that $\boldsymbol{x}^\star(\boldsymbol{q}_\kappa, \boldsymbol{\rho}_\kappa)$ is the most likely solution to the parameter $\hat{\boldsymbol{\rho}}_\kappa = \mathcal{M}_\theta(\phi_\kappa)$.

## 4 Methodology

We start by defining a function $UNSAT_i(\boldsymbol{x}, \boldsymbol{\rho}_\kappa)$ to indicate whether the $i$-th constraint is violated by an assignment $\boldsymbol{x}$ when the constraint parameter is set to the vector $\boldsymbol{\rho}_\kappa$. It is defined as:

$$UNSAT_i(\boldsymbol{x}, \boldsymbol{\rho}_\kappa) := \mathbb{1}\{g_i(\boldsymbol{x}, \boldsymbol{\rho}_\kappa) > 0\}$$

Similarly, we define $SAT_i(\boldsymbol{x}, \boldsymbol{\rho}_\kappa)$ to indicate that the $i$-th constraint is satisfied. Therefore, $SAT_i(\boldsymbol{x}, \boldsymbol{\rho}_\kappa) := 1 - UNSAT_i(\boldsymbol{x}, \boldsymbol{\rho}_\kappa)$. Using these definitions, we introduce $Feas(\boldsymbol{x}, \boldsymbol{\rho}_\kappa)$, an indicator variable equal to 1 if the solution $\boldsymbol{x}$ satisfies all the constraints.

$$Feas(\boldsymbol{x}, \boldsymbol{\rho}_\kappa) = \prod_{i \in \mathcal{S}} SAT_i(\boldsymbol{x}, \boldsymbol{\rho}_\kappa) \tag{4}$$

We use $Infeas(\boldsymbol{x}, \boldsymbol{\rho}_\kappa)$ to denote that $\boldsymbol{x}$ is infeasible with respect to $\boldsymbol{\rho}_\kappa$. Note that $\boldsymbol{x}$ is an infeasible assignment if it violates any one of the constraints. Hence, we can write:

$$Infeas(\boldsymbol{x}, \boldsymbol{\rho}_\kappa) = \max_{i \in \mathcal{S}} UNSAT_i(\boldsymbol{x}, \boldsymbol{\rho}_\kappa) \tag{5}$$

As we want to maximize the likelihood of obtaining a feasible solution with optimal objective value, we first model the probability of violating the constraints using the predicted parameters. Given predicted parameters $\mathcal{M}_\theta(\boldsymbol{\phi}_\kappa)$, we model the parametric probability $P_\theta\big(UNSAT_i(\boldsymbol{x}, \mathcal{M}_\theta(\boldsymbol{\phi}_\kappa))\big|\mathcal{M}_\theta(\boldsymbol{\phi}_\kappa)\big)$ in the following manner:

$$P_\theta\Big(UNSAT_i\big(\boldsymbol{x}, \mathcal{M}_\theta(\boldsymbol{\phi}_\kappa)\big)\Big|\mathcal{M}_\theta(\boldsymbol{\phi}_\kappa)\Big) = \frac{1}{1 + \exp(-g_i(\boldsymbol{x}, \mathcal{M}_\theta(\boldsymbol{\phi}_\kappa)))} \tag{6}$$

Note that the above is a smooth version of the following non-smooth distribution:

$$P_\theta = \begin{cases} 1, & \text{if } g_i(\boldsymbol{x}, \mathcal{M}_\theta(\boldsymbol{\phi}_\kappa)) \geq 0 \\ 0, & \text{otherwise} \end{cases} \tag{7}$$

As $P_\theta(SAT_i(\boldsymbol{x}, \mathcal{M}_\theta(\boldsymbol{\phi}_\kappa))|\mathcal{M}_\theta(\boldsymbol{\phi}_\kappa)) = 1 - P_\theta(UNSAT_i(\boldsymbol{x}, \mathcal{M}_\theta(\boldsymbol{\phi}_\kappa))|\mathcal{M}_\theta(\boldsymbol{\phi}_\kappa))$, we can write:

$$P_\theta\Big(SAT_i(\boldsymbol{x}, \mathcal{M}_\theta(\boldsymbol{\phi}_\kappa))\Big|\mathcal{M}_\theta(\boldsymbol{\phi}_\kappa)\Big) = \frac{1}{1 + \exp(g_i(\boldsymbol{x}, \mathcal{M}_\theta(\boldsymbol{\phi}_\kappa)))} \tag{8}$$

### 4.1 Loss Function to Penalize Infeasibility-Inducing Predictions

We first propose a loss function that ensures that true infeasible assignments remain infeasible under the predicted parameters. Let $\boldsymbol{x}_{neg}$ be an assignment that is infeasible under the true parameters. As it is easy to determine which constraints are violated by $\boldsymbol{x}_{neg}$, we develop an MLE framework to maximize the likelihood of violating those constraints under the predicted parameters. Given a $\boldsymbol{x}_{neg}$, the log-likelihood of violating these constraints can be written in the following form:

$$\sum_{i \in \mathcal{S}} UNSAT_i(\boldsymbol{x}_{neg}, \boldsymbol{\rho}_\kappa) \log P_\theta\Big(UNSAT_i(\boldsymbol{x}_{neg}, \mathcal{M}_\theta(\boldsymbol{\phi}_\kappa))\Big|\mathcal{M}_\theta(\boldsymbol{\phi}_\kappa)\Big) \tag{9}$$

Note that $UNSAT_i(\boldsymbol{x}_{neg}, \boldsymbol{\rho}_\kappa)$ can be viewed as a mask that equals 1 when $\boldsymbol{x}_{neg}$ violates constraint $i$ under the true parameters $\boldsymbol{\rho}_\kappa$. Using the definition of $P_\theta(UNSAT_i(\boldsymbol{x}, \mathcal{M}_\theta(\boldsymbol{\phi}_\kappa))|\mathcal{M}_\theta(\boldsymbol{\phi}_\kappa))$ from Eq. 6, we can write the following:

$$\log P_\theta\Big(UNSAT_i(\boldsymbol{x}_{neg}, \mathcal{M}_\theta(\boldsymbol{\phi}_\kappa))\Big|\mathcal{M}_\theta(\boldsymbol{\phi}_\kappa)\Big) = -\log(1 + \exp(-g_i(\boldsymbol{x}_{neg}, \mathcal{M}_\theta(\boldsymbol{\phi}_\kappa)))) \tag{10}$$

We now propose a loss function to maximize this likelihood (i.e., minimize the negative log-likelihood). We include a margin parameter $\Upsilon > 0$ to encourage the constraints to be violated with a buffer. Since this loss penalizes cases where an infeasible assignment $\boldsymbol{x}_{neg}$ is considered feasible under the predicted parameters, we refer to it as the Infeasibility-Aware Loss (IAL).

$$\mathcal{L}^{IAL}(\boldsymbol{x}_{neg}, \mathcal{M}_\theta(\boldsymbol{\phi}_\kappa)) = \sum_{i \in \mathcal{S}} UNSAT_i(\boldsymbol{x}_{neg}, \boldsymbol{\rho}_\kappa)\, softplus\big(\Upsilon - g_i(\boldsymbol{x}_{neg}, \mathcal{M}_\theta(\boldsymbol{\phi}_\kappa))\big) \tag{11}$$

where we write $\log(1 + \exp(.))$ as $softplus(.)$.

**Intuition behind IAL.** Note that $softplus$ is a smooth approximation to the $Relu$ function. If we replace the $softplus$ function with a $Relu$ function, Eq. 11 takes the following form:

$$\overline{\mathcal{L}^{IAL}}(\boldsymbol{x}_{neg}, \mathcal{M}_\theta(\boldsymbol{\phi}_\kappa)) = \sum_{i \in \mathcal{S}} UNSAT_i(\boldsymbol{x}_{neg}, \boldsymbol{\rho}_\kappa) \max\big(0, \Upsilon - g_i(\boldsymbol{x}_{neg}, \mathcal{M}_\theta(\boldsymbol{\phi}_\kappa))\big) \tag{12}$$

From this expression, we see that for a truly violated constraint, i.e., of which $UNSAT_i(\boldsymbol{x}_{neg}, \boldsymbol{\rho}_\kappa) = 1$, the loss becomes zero only if $g_i(\boldsymbol{x}_{neg}, \mathcal{M}_\theta(\boldsymbol{\phi}_\kappa)) > \Upsilon$. In other words, the predicted constraint must also be violated. $\overline{\mathcal{L}^{IAL}}(\boldsymbol{x}_{neg}, \mathcal{M}_\theta(\boldsymbol{\phi}_\kappa))$ is zero if this holds for all constraints violated by $\boldsymbol{x}_{neg}$.

**Selecting $x_{neg}$.** The IAL loss formulation is based on $x_{neg}$, a candidate assignment which violates at least one constraint with respect to the true parameter $\rho_\kappa$. This leads to the question of how to obtain such a $x_{neg}$. We adopt the following approach: for each instance, after predicting $\hat{\rho}_\kappa$, we solve the COP using $\hat{\rho}_\kappa$ and obtain a solution $x^\star(q_\kappa, \hat{\rho}_\kappa)$. If this solution is infeasible under the true parameters $\rho_\kappa$, we use it as $x_{neg}$. Otherwise, we set the loss to zero, since the predicted solution satisfies all the true constraints, which is the preferred outcome. We call this final loss function the Infeasibility Penalty Loss (IPL), as it penalizes the predicted parameters when the resulting predicted solution violates the true constraints.

$$\mathcal{L}^{IPL}(\rho_\kappa, \hat{\rho}_\kappa) = (1 - Feas\left(x^\star\left(q_\kappa, \hat{\rho}_\kappa\right), \rho_\kappa\right)) \mathcal{L}^{IAL}(x^\star(q_\kappa, \hat{\rho}_\kappa), \hat{\rho}_\kappa) \tag{13}$$

We can also define $\overline{\mathcal{L}^{IPL}}$ where $\mathcal{L}^{IAL}$ in Eq. 13 is replaced with $\overline{\mathcal{L}^{IAL}}$.

**Lemma 1.** *For a true parameter $\rho_\kappa$ and a predicted parameter $\hat{\rho}_\kappa$, $\overline{\mathcal{L}^{IPL}}(\rho_\kappa, \hat{\rho}_\kappa)$ is zero if and only if $x^\star\left(q_\kappa, \hat{\rho}_\kappa\right)$, an optimal solution for $\hat{\rho}_\kappa$, is feasible with respect to $\rho_\kappa$.*

*Proof.* $\overline{\mathcal{L}^{IPL}}$ is by construction zero when $x^\star\left(q_\kappa, \hat{\rho}_\kappa\right)$ is feasible with respect to $\rho_\kappa$. Because if $x^\star\left(q_\kappa, \hat{\rho}_\kappa\right)$ is feasible with respect to $\rho_\kappa$, $Feas\left(x^\star\left(q_\kappa, \hat{\rho}_\kappa\right), \rho_\kappa\right)$ is equal to one, which makes $\overline{\mathcal{L}^{IPL}} = 0$. Hence,

$$Feas\left(x^\star\left(q_\kappa, \hat{\rho}_\kappa\right), \rho_\kappa\right) = 1 \implies \overline{\mathcal{L}^{IPL}} = 0$$

To prove the other direction, let us assume, $\overline{\mathcal{L}^{IPL}} = 0$, but $Feas\left(x^\star\left(q_\kappa, \hat{\rho}_\kappa\right), \rho_\kappa\right) \neq 1$. Hence, $\overline{\mathcal{L}^{IAL}}(x^\star(q_\kappa, \hat{\rho}_\kappa), \hat{\rho}_\kappa)$ must be zero. Note that, $\overline{\mathcal{L}^{IAL}}(x^\star(q_\kappa, \hat{\rho}_\kappa), \hat{\rho}_\kappa)$ can be zero only if $g_i(x^\star(q_\kappa, \hat{\rho}_\kappa), \hat{\rho}_\kappa) \geq \Upsilon > 0$ for all the violated constraints. However, as $x^\star(q_\kappa, \hat{\rho}_\kappa), \hat{\rho}_\kappa)$ is the optimal solution with $\hat{\rho}_\kappa$ being the parameter, $g_i(x^\star(q_\kappa, \hat{\rho}_\kappa), \hat{\rho}_\kappa)$ must be less than zero for all the constraints and we arrived at a contradiction. So,

$$\overline{\mathcal{L}^{IPL}} = 0 \implies Feas\left(x^\star\left(q_\kappa, \hat{\rho}_\kappa\right), \rho_\kappa\right) = 1$$

$\square$

### 4.2 Loss Function for Preserving Optimal Solutions

The IPL loss penalizes predicted parameters when the predicted solution becomes infeasible with respect to the true parameters. However, minimizing *only* $\mathcal{L}^{IPL}$ can lead to learning a feasible region that is much stricter than the true feasible region, as any solution therein will be feasible under the true parameters. For example, when learning the capacity parameter in a knapsack problem, minimizing only $\mathcal{L}^{IPL}$ might result in capacity values so small that no items are selected. While such solutions are always feasible with respect to the true parameters, they are suboptimal and of no practical value. Therefore, we also need a loss function to penalize cases where the learned parameters turn out to be too strict. To do this, we define a loss function that penalizes predicted parameters when they render known optimal solutions infeasible. Within the MLE framework, this corresponds to maximizing the likelihood that the true solution is feasible.

Thus, we formulate the probability of an assignment $x$ being a feasible point, given the parameter $\mathcal{M}_\theta(\phi_\kappa)$. Note that $x$ is a feasible assignment only if it satisfies all the constraints. While the probabilities of satisfying constraints $i$ and $j$ are *not* independent, they are conditionally independent [7, Chapter 8] given the constraint parameter vector $\mathcal{M}_\theta(\phi_\kappa)$. This is because the constraint parameter vector determines the feasibility of each constraint independently. Once $\mathcal{M}_\theta(\phi_\kappa)$ is known, the probability of satisfying constraint $i$ depends only on $\mathcal{M}_\theta(\phi_\kappa)$, and any additional information about constraint $j$ being satisfied does not affect that probability. Because of the conditional independence, we can express the conditional probability of the assignment, $x$ being feasible given $\mathcal{M}_\theta(\phi_\kappa)$, by:

$$P_\theta\left(Feas\left(x, \mathcal{M}_\theta(\phi_\kappa)\right) \middle| \mathcal{M}_\theta(\phi_\kappa)\right) = \prod_{i \in \mathcal{S}} P_\theta\left(SAT_i\left(x, \mathcal{M}_\theta(\phi_\kappa)\right) | \mathcal{M}_\theta(\phi_\kappa)\right) \tag{14}$$

Now using Eq. 8, we can model $P_\theta\left(Feas\left(x, \mathcal{M}_\theta(\phi_\kappa) | \mathcal{M}_\theta(\phi_\kappa)\right)\right)$, as

$$P_\theta\left(Feas\left(x, \mathcal{M}_\theta(\phi_\kappa)\right) \middle| \mathcal{M}_\theta(\phi_\kappa)\right) = \prod_{i \in \mathcal{S}} \frac{1}{1 + \exp(g_i(x, \mathcal{M}_\theta(\phi_\kappa)))} \tag{15}$$

Thus, we obtain the log-likelihood function of $\boldsymbol{x}$ being feasible, in the form of:

$$-\sum_{i\in\mathcal{S}}\log\left(1+\exp\left(g_i\left(\boldsymbol{x},\mathcal{M}_\theta(\boldsymbol{\phi}_\kappa)\right)\right)\right)=-\sum_{i\in\mathcal{S}}softplus\left(g_i(\boldsymbol{x}_\kappa^\star,\mathcal{M}_\theta(\boldsymbol{\phi}_\kappa))\right) \qquad (16)$$

We point out that in the training data, we know that $\boldsymbol{x}^\star(\boldsymbol{q}_\kappa,\boldsymbol{\rho}_\kappa)$ is a feasible solution. Denoting $\boldsymbol{x}_\kappa^\star$ as the known feasible point, we can formulate a loss function by minimizing the negative log-likelihood. In our implementation, we add a margin parameter $\Upsilon$ as we have done for $\mathcal{L}^{IPL}$. Since this loss is designed to maximize the probability that the optimal solution remains feasible, we refer to it as the Optimality-Preserving Loss (OPL). We express it in the following form:

$$\mathcal{L}^{OPL}(\boldsymbol{\rho}_\kappa,\mathcal{M}_\theta(\boldsymbol{\phi}_\kappa))=\sum_{i\in\mathcal{S}}softplus\left(\Upsilon+g_i\left(\boldsymbol{x}^\star\left(\boldsymbol{q}_\kappa,\boldsymbol{\rho}_\kappa\right),\mathcal{M}_\theta(\boldsymbol{\phi}_\kappa)\right)\right) \qquad (17)$$

**Intuition behind OPL.** By replacing the *softplus* in Eq. 17 with a *Relu* function, we can write

$$\overline{\mathcal{L}^{OPL}}(\boldsymbol{\rho}_\kappa,\hat{\boldsymbol{\rho}}_\kappa)=\sum_{i\in\mathcal{S}}\max\left(\left(\Upsilon+g_i\left(\boldsymbol{x}^\star\left(\boldsymbol{q}_\kappa,\boldsymbol{\rho}_\kappa\right),\hat{\boldsymbol{\rho}}_\kappa\right)\right),0\right) \qquad (18)$$

where we write $\mathcal{M}_\theta(\boldsymbol{\phi}_\kappa)$ as $\hat{\boldsymbol{\rho}}_\kappa$ for brevity. From this expression, it is easy to see that $\mathcal{L}^{OPL}$ attains its minimum value of zero when all constraint values are less than or equal to $-\Upsilon$, i.e., when all the constraints are satisfied with some margin.

**Lemma 2.** *For a true parameter $\boldsymbol{\rho}_\kappa$ and a predicted parameter $\hat{\boldsymbol{\rho}}_\kappa$, both $\overline{\mathcal{L}^{OPL}}(\boldsymbol{\rho}_\kappa,\hat{\boldsymbol{\rho}}_\kappa)$ and $\overline{\mathcal{L}^{IPL}}(\boldsymbol{\rho}_\kappa,\hat{\boldsymbol{\rho}}_\kappa)$ will be zero if and only if $\boldsymbol{x}^\star\left(\boldsymbol{q}_\kappa,\hat{\boldsymbol{\rho}}_\kappa\right)$, a solution of $\hat{\boldsymbol{\rho}}_\kappa$, is an optimal solution with respect to $\boldsymbol{\rho}_\kappa$.*

*Proof.* Let $\overline{\mathcal{L}^{OPL}}(\boldsymbol{\rho}_\kappa,\hat{\boldsymbol{\rho}}_\kappa)$ and $\overline{\mathcal{L}^{IPL}}(\boldsymbol{\rho}_\kappa,\hat{\boldsymbol{\rho}}_\kappa)$ be equal to zero. We have proved in Lemma 1 that $\overline{\mathcal{L}^{IPL}}$ being zero implies that a solution of $\hat{\boldsymbol{\rho}}_\kappa$ is feasible with respect to $\kappa$. So, $\boldsymbol{x}^\star\left(\boldsymbol{q}_\kappa,\hat{\boldsymbol{\rho}}_\kappa\right)$, a solution of $\hat{\boldsymbol{\rho}}_\kappa$ lies in the set $\mathcal{F}(\boldsymbol{\rho}_\kappa)$.

Because $\boldsymbol{x}^\star(\boldsymbol{q}_\kappa,\boldsymbol{\rho}_\kappa)$ is an optimal solution,

$$f(\boldsymbol{x}^\star(\boldsymbol{q}_\kappa,\boldsymbol{\rho}_\kappa);\boldsymbol{q}_\kappa)\leq f(\boldsymbol{x}';\boldsymbol{q}_\kappa)\quad\forall\boldsymbol{x}'\in\mathcal{F}(\boldsymbol{\rho}_\kappa)$$

As $\boldsymbol{x}^\star\left(\boldsymbol{q}_\kappa,\hat{\boldsymbol{\rho}}_\kappa\right)\in\mathcal{F}(\boldsymbol{\rho}_\kappa)$, from the above argument, we can write

$$f(\boldsymbol{x}^\star(\boldsymbol{q}_\kappa,\boldsymbol{\rho}_\kappa);\boldsymbol{q}_\kappa)\leq f(\boldsymbol{x}^\star\left(\boldsymbol{q}_\kappa,\hat{\boldsymbol{\rho}}_\kappa\right);\boldsymbol{q}_\kappa) \qquad (19)$$

Now, $\overline{\mathcal{L}^{OPL}}(\boldsymbol{\rho}_\kappa,\hat{\boldsymbol{\rho}}_\kappa)$ being zero implies that $\boldsymbol{x}^\star(\boldsymbol{q}_\kappa,\boldsymbol{\rho}_\kappa)$, an optimal solution of the true parameter, $\boldsymbol{\rho}_\kappa$ lies in $\mathcal{F}(\hat{\boldsymbol{\rho}}_\kappa)$. Note,

$$f(\boldsymbol{x}^\star(\boldsymbol{q}_\kappa,\hat{\boldsymbol{\rho}}_\kappa);\boldsymbol{q}_\kappa)\leq f(\boldsymbol{x}';\boldsymbol{q}_\kappa)\quad\forall\boldsymbol{x}'\in\mathcal{F}(\hat{\boldsymbol{\rho}}_\kappa)$$

As $\boldsymbol{x}^\star(\boldsymbol{q}_\kappa,\boldsymbol{\rho}_\kappa)\in\mathcal{F}(\hat{\boldsymbol{\rho}}_\kappa)$

$$f(\boldsymbol{x}^\star(\boldsymbol{q}_\kappa,\hat{\boldsymbol{\rho}}_\kappa);\boldsymbol{q}_\kappa)\leq f(\boldsymbol{x}^\star\left(\boldsymbol{q}_\kappa,\boldsymbol{\rho}_\kappa\right);\boldsymbol{q}_\kappa) \qquad (20)$$

Hence, Eq. 19 and Eq. 20 suggests

$$f(\boldsymbol{x}^\star(\boldsymbol{q}_\kappa,\hat{\boldsymbol{\rho}}_\kappa);\boldsymbol{q}_\kappa)=f(\boldsymbol{x}^\star\left(\boldsymbol{q}_\kappa,\boldsymbol{\rho}_\kappa\right);\boldsymbol{q}_\kappa)$$

As $\boldsymbol{x}^\star(\boldsymbol{q}_\kappa,\hat{\boldsymbol{\rho}}_\kappa)$ is a feasible solution with respect to $\boldsymbol{\rho}_\kappa$ and its objective value is equal to the optimal objective value, it is an optimal solution.

Now, to prove the opposite direction, assume that $\boldsymbol{x}^\star\left(\boldsymbol{q}_\kappa,\hat{\boldsymbol{\rho}}_\kappa\right)$ is an optimal solution with respect to $\boldsymbol{\rho}_\kappa$. This implies that $\boldsymbol{x}^\star\left(\boldsymbol{q}_\kappa,\hat{\boldsymbol{\rho}}_\kappa\right)$ is a feasible solution with respect to $\boldsymbol{\rho}_\kappa$. Therefore, from Lemma 1, $\overline{\mathcal{L}^{IPL}}(\boldsymbol{\rho}_\kappa,\hat{\boldsymbol{\rho}}_\kappa)=0$. As $\boldsymbol{x}^\star\left(\boldsymbol{q}_\kappa,\hat{\boldsymbol{\rho}}_\kappa\right)$ adheres to all the constraints when $\hat{\boldsymbol{\rho}}_\kappa$ is the constraints parameter value,

$$\sum_{i\in\mathcal{S}}\max\left(\left(\Upsilon+g_i(\boldsymbol{x}^\star\left(\boldsymbol{q}_\kappa,\hat{\boldsymbol{\rho}}_\kappa\right),\hat{\boldsymbol{\rho}}_\kappa)\right),0\right)=0 \qquad (21)$$

As $\boldsymbol{x}^\star\left(\boldsymbol{q}_\kappa,\hat{\boldsymbol{\rho}}_\kappa\right)$ is an optimal solution with respect to $\boldsymbol{\rho}_\kappa$, $\boldsymbol{x}^\star\left(\boldsymbol{q}_\kappa,\hat{\boldsymbol{\rho}}_\kappa\right)$ is one of the $\boldsymbol{x}^\star\left(\boldsymbol{q}_\kappa,\boldsymbol{\rho}_\kappa\right)$. If we replace $\boldsymbol{x}^\star\left(\boldsymbol{q}_\kappa,\hat{\boldsymbol{\rho}}_\kappa\right)$ with $\boldsymbol{x}^\star\left(\boldsymbol{q}_\kappa,\boldsymbol{\rho}_\kappa\right)$ in Eq. 21, $\overline{\mathcal{L}^{OPL}}(\boldsymbol{\rho}_\kappa,\hat{\boldsymbol{\rho}}_\kappa)=0$ by definition (Eq. 18). $\square$

### 4.3 Minimizing a Convex Combination of $\mathcal{L}^{OPL}$ and $\mathcal{L}^{IPL}$

Lemma 2 suggests that by minimizing both $\mathcal{L}^{IPL}$ and $\mathcal{L}^{OPL}$ to zero, we can obtain a solution $x^\star(q_\kappa, \hat{\rho}_\kappa)$, which is optimal with respect to $\rho_\kappa$. However, simultaneously minimizing $\mathcal{L}^{IPL}$ and $\mathcal{L}^{OPL}$ presents a challenge. This is because $\mathcal{L}^{IPL}$ and $\mathcal{L}^{OPL}$ represent two conflicting goals. Minimizing only $\mathcal{L}^{OPL}$ encourages the model to predict loose constraints so that all true feasible solutions remain feasible, hence making even infeasible solutions appear feasible. Conversely, minimizing $\mathcal{L}^{IPL}$ encourages stricter constraints, which may render even true optimal solutions infeasible. We thus propose to minimize a convex combination of these losses, in the following form:

$$\alpha \mathcal{L}^{IPL}(\rho_\kappa, \hat{\rho}_\kappa) + (1 - \alpha)\mathcal{L}^{OPL}(\rho_\kappa, \hat{\rho}_\kappa) \tag{22}$$

$\alpha \in [0, 1]$ is a hyperparameter we call the *infeasibility-aversion coefficient*. It reflects how much importance the decision-maker places on avoiding infeasible solutions (through IPL) versus maintaining the feasibility of known feasible ones (through OPL). The two extremes are $\alpha = 1$ and $\alpha = 0$, both of which represent unrealistic preferences. $\alpha = 1$ means the decision-maker prioritizes only feasibility, possibly ending up with trivial solutions (e.g., picking no items in a knapsack problem). $\alpha$ near 0 implies a focus on high-reward solutions, even if they are infeasible (e.g., selecting all items).

In this section, we have proposed a novel approach for 'optimizing decisions through end-to-end constraint estimation' (Odece), involving two novel loss functions: $\mathcal{L}^{IPL}$ and $\mathcal{L}^{OPL}$. We have proposed to consider a weighted average of the two losses to reflect the decision-maker's subjective preference of infeasibility over suboptimality. Next, we will experimentally investigate how Odece performs compared to other existing approaches for predicting constraint parameters.

## 5 Experimental Evaluation

We experiment with predicting constraint parameters in the following optimization problems:

**Multi-dimensional knapsack problem.** The first is the 0-1 multi-dimensional knapsack problem (MDKP), introduced in Section 3. In the experimental section, we consider MDKP instances with 50 items and constraints in three dimensions. We consider two settings: 1) predicting the item weight vectors while keeping the capacity constraints known, and 2) predicting the capacity vectors with known item weights.

**Brass Alloy Production.** Our third experimental setting involves the Brass alloy production problem. We reproduce this experimental setup from the work by Hu et al. [15]. It is a covering LP problem. We use the publicly available Brass alloy data from their repository.[2] The production of the Brass alloy requires two metals: Copper (Cu) and Zinc (Zn). A factory needs to purchase ores that contain these two metals. There are 10 potential suppliers, each offering ores at a different price per unit. The exact metal content in each supplier's ore is unknown and must be predicted. The goal is to buy a combination of ores from the 10 suppliers at the lowest possible cost, while ensuring that the total purchase contains at least 627.54 units of Cu and 369.72 units of Zn.

### 5.1 Dataset Description

**Synthetic parameter generation of the MDKP.** We adopt the synthetic data generation process described by Elmachtoub and Grigas [12]. This data generation approach introduces a non-linear relationship between input features and the unknown parameters. A linear model is used to predict the parameters from features. The motivation behind using a linear model is to demonstrate that DFL methods can produce high-quality solutions despite model misspecification. This approach has been widely used as a benchmark in DFL studies [12, 20, 22, 30, 31].

Each knapsack instance contains 50 items, with all items correlated with 10 input features. Item values, which appear in the objective function, are sampled independently from a Gumbel distribution with location 100 and scale 20. For each run, 1500 instances are generated, split into 900 for training, 100 for validation, and 500 for testing. We took extra caution to ensure that the synthetically-generated parameters produce non-trivial optimal solutions, avoiding cases where all or none of the items are

---

[2]`https://github.com/Elizabethxyhu/NeurIPS_Two_Stage_Predict-Optimize/`

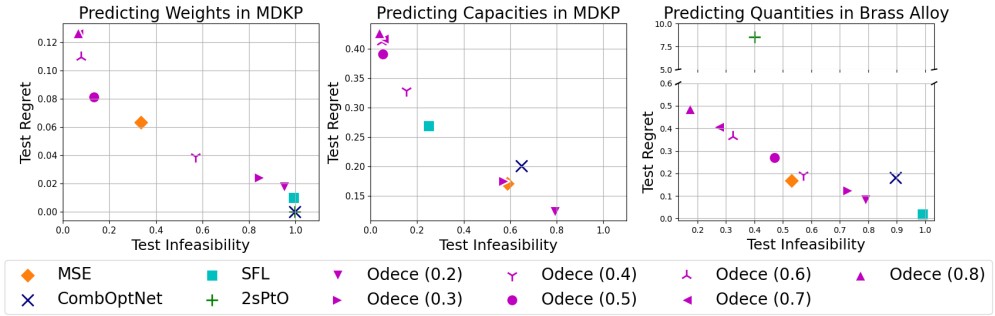

Figure 1: Infeasibility ratio and regret of feasible solutions on Test instances (best viewed in colors).

selected in the knapsack. To achieve this, we clip the capacity in each dimension to be less than half the total weight of all items and ensure that individual item weights remain below the respective capacity values.

**Brass Alloy Production.**  For this experiment, we use the dataset from Hu et al. [15]. In this dataset, the Cu and Zn contents in each supplier's ore are predicted using separate sets of 4096 features each. Out of the available 500 instances, we use 350, 50, and 100 for training, validation, and testing, respectively. We use a fully-connected neural network with one hidden layer of 512 neurons for prediction.

The experiments were executed on an *Intel i7-13800H* (20 cores) CPU with 32GB RAM. Details on the problems and datasets used in the three experiments are provided in Appendix A. The source code for reproducing our results is publicly available at: `https://github.com/JayMan91/OdeceDFLforConstraintsNeurips25`.

## 5.2   Experimental Results

We denote the model as Odece($\alpha$) (e.g., Odece(0.1)) to indicate Odece is trained with that specific value of $\alpha$. We compare the proposed Odece against four competitors: i) *MSE*: a PFL approach which trains the ML model to minimize the MSE loss over the parameter predictions, ii) *CombOptNet*: the technique proposed by Paulus et al. [26] to compute the gradient of ILP parameters, iii) *SFL*: the solver-free learning proposed by Nandwani et al. [24] for learning parameters of an ILP,  iv) *2sPtO*: the two-stage predict+optimize approach proposed by Hu et al. [15]. We implement 2sPtO using a high penalty factor, as recommended in Appendix A.2 of their paper. We could not apply 2sPtO to predict the knapsack capacity, as it is designed to predict only the left-hand side parameters in linear constraints. More specifically, it can compute gradients with respect to $b$ in constraints of the form $b^\top x \leq c$, but not with respect to $c$. For CombOptNet we compute the L1 loss of the predicted solution and the true solution and backpropagate the L1 loss through CombOptNet.

We evaluate performance using two metrics: the proportion of infeasible solutions and the normalized regret on the test data. Each technique is run five times with different random seeds. Figure 1 reports the average proportion of infeasible solutions (x-axis) and average normalized regret (y-axis) across these runs. Detailed results, including averages and standard deviations across the five runs, are provided in Appendix B. We compute regret only over predicted solutions that are feasible under the true parameters. More formally, out of $K$ test instances, let $K'$ denote the number of instances where the solution $x^\star(q_\kappa, \hat{\rho}_\kappa)$ is feasible under $\rho_\kappa$. The proportion of infeasible solutions is then $\frac{K'}{K}$. The normalized regret is defined as follows:

$$\frac{1}{K'} \sum_{\kappa=1}^{K'} \frac{f\left(x^\star\left(q_\kappa, \hat{\rho}_\kappa\right); q_\kappa\right) - f(x_\kappa^\star; q_\kappa)}{f(x_\kappa^\star; q_\kappa)} \tag{23}$$

Note that normalized regret can be misleading in this setting. This is because in extreme cases, where most predicted solutions are infeasible, regret is computed over only a small subset of instances.

**Results.**   The first observation from Figure 1 is that CombOptNet exhibits relatively poor performance across the three problems. For both MDKP weight prediction and the Brass Alloy problem, the proportion of infeasible solutions generated by CombOptNet is very high ($> 85\%$). In MDKP capacity prediction, the proportion of infeasible solutions is slightly lower but still high ($> 65\%$). The performance of SFL is similar to CombOptNet for MDKP weight prediction and the Brass Alloy problem. However, in the MDKP capacity prediction problem, SFL performs significantly better, with the proportion of infeasible solutions around $25\%$. We suspect this is because SFL relies more on $c$ than $b$ to separate the true optimal solution from the *negative* assignments, resulting in more informative gradients for $c$ and less informative gradients for $b$. (For example, in a 3-dimensional MDKP where the optimal solution is $[0, 0, 1]$, negative samples such as $[1, 1, 1], [1, 1, 0], [0, 1, 1]$ and $[1, 0, 1]$ can be classified as negative based on the capacity, i.e., the right-hand side parameter, $c$.)

For MDKP weight prediction, 2sPtO performs very poorly (the proportion of infeasible solutions is greater than $99\%$). It is not applicable for MDKP capacity prediction. Figure 1 might suggest that 2sPtO achieves a relatively lower proportion of infeasible solutions for the Brass Alloy problem, but this is misleading for two reasons. First, it results in extremely high normalized regret ($> 850\%$). Note that we have to break the y-axis in Figure 1 to accommodate 2sPtO's regret values. More importantly, its performance is highly unstable, which is not noticeable in Figure 1, where the average of five runs is plotted. Table 3 in the Appendix reveals that the standard deviation across the five runs is very high. In some runs, the proportion of infeasible solutions is near 0, while in others it reaches $\approx 99\%$. This indicates unstable learning.

Our closest competitor is MSE, but minimizing MSE does not let decision-makers adjust the trade-off between optimality and feasibility. MSE has infeasibility rates of 33%, 60%, and 53% across the three problems, respectively. For a decision-maker, these infeasibility rates may be unacceptably high. Odece allows a decision-maker to attain lower infeasibility rate by varying $\alpha$, at the cost of higher normalized regret. For instance, Odece(0.8) brings the infeasibility rates down to 6%, 4%, and 18%, across the three problems, respectively. We have also conducted statistical significance tests to determine whether the test regret and infeasibility of Odece are lower than those of MSE. Since each run was conducted using the same random seed for both models, we used a paired t-test with the alternative hypothesis that MSE has higher regret and higher infeasibility than Odece. The p-values from the paired t-tests for different values of $\alpha$ are reported in Table 4, 5 and 6 in Appendix B. The results of the statistical significance test reveal that for $\alpha > 0.5$, Odece produces a statistically significantly lower proportion of infeasible solutions than MSE. On the other hand, for $\alpha < 0.3$, the normalized regret on feasible instances generated by Odece is statistically significantly lower than that of MSE. This holds true across all three problems.

## 6   Conclusion

We proposed Odece, a novel DFL technique for predicting parameters appearing in the constraints of COPs. Unlike existing DFL methods for constraint predictions – which either depend on problem-specific second-stage corrective actions or which are designed to predict the full set of constraints – Odece directly optimizes for feasibility and solution quality in a single stage. Our experiments showed that by tuning Odece's $\alpha$ parameter, decision-makers can flexibly manage the trade-off between infeasibility and suboptimality, allowing them to align solutions with their individual preferences.

One limitation of the current Odece implementation is that it solves the COP for each instance using the predicted parameters during training. To speed up training, future work could explore caching candidate assignments, as done by Mulamba et al. [23] for predicting objective parameters. Odece could also be extended to jointly predict both constraint and objective parameters. Finally, we plan to apply Odece to a wider range of optimization problems, such as combinatorial problems with non-linear objective and constraint functions, as well as larger, real-world applications.

## Acknowledgments and Disclosure of Funding

This research received funding from the European Research Council (ERC) under the European Union's Horizon 2020 Research and Innovation Programme (Grant No. 101002802, CHAT-Opt), and from the Research Foundation Flanders (FWO) project G0G3220N. Senne Berden is a fellow of the Research Foundation-Flanders (FWO-Vlaanderen, 11PQ024N).

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

## A  Problem and Dataset Description

### A.1  Multi-dimensional knapsack problem

The formulation of the optimization problem is as follows:

$$\min_{x_{1:N}} \sum_{n=1}^{N} (-q_n) x_n \quad \text{such that} \quad x_n \in \{0, 1\}, \ \forall n \in [N], \quad \sum_{n=1}^{N} \rho_{ni} x_n \le \rho_i, \ \forall i \in [M] \tag{24}$$

Here, $q_n$ and $\rho_{ni}$ are the values and weights in dimension $i$ of item $n$. We synthetically generate 1500 MDKP instances for each run. Of these, 900 are used for training, 100 for validation, and 500 for testing. We represent the dataset as $\{(\phi_\kappa, \rho_\kappa, q_\kappa, x^\star(q_\kappa, \rho_\kappa))\}_{\kappa=1}^{K}$ where $\phi_\kappa$ is the feature vector, $\rho_\kappa$ is the concatenated vector of weights and capacity, $q_\kappa$ is the objective parameter, and $x^\star(q_\kappa, \rho_\kappa)$ denotes the solution to instance $\kappa$. In our experiment, $\phi_\kappa$ is of dimension 10. The feature vectors are sampled from a multivariate Gaussian distribution with zero mean and unit variance, i.e., $\phi_\kappa \sim \mathbf{N}(0, \mathbf{I}_{10})$.

#### A.1.1  Data Generation for Unknown Weight Vectors

We generate our dataset similarly to Elmachtoub and Grigas [12] and Tang and Khalil [30]. While these works consider predicting parameters in the objective function, we focus on predicting parameters in the constraints. We use the same data generation process to synthetically create constraint parameters from features.

To generate the weight vector, first a matrix $B \in \mathbb{R}^{M \times N \times 10}$ is generated, which represents the true underlying model, unknown to the decision-maker. Each entry of the matrix $B$ is sampled from independent Bernoulli distributions of probability 0.5. The weight $\rho_{ni}^{(\kappa)}$, for $\kappa$-th instance, is then generated according to the following formula:

$$\rho_{ni}^{(\kappa)} = \left[ \frac{1}{3.5^{\text{Deg}}} \left( \frac{1}{\sqrt{10}} \left( \phi_\kappa^\top B[i, n] \right) + 3 \right)^{\text{Deg}} + 1 \right] \xi_{ni}^{(\kappa)} \tag{25}$$

The *Deg* is 'model misspecification' parameter. This is because a linear model is used as a predictive model in the experiment and a higher value of *Deg* indicates the predictive model deviates more from the true underlying model and larger the prediction errors. $\xi_{ni}^{(\kappa)}$ is a multiplicative noise term sampled randomly from the uniform distribution $\xi_{ni}^{(\kappa)} \sim U[1 - w, 1 + w]$. We report results for *Deg* = 6 and $w = 0.25$.

We generated the capacity $\rho_i^{(\kappa)}$, the the following ways:

$$\rho_i^{(\kappa)} = r * \xi_i^{(\kappa)} \left[ \sum_{n=1}^{N} \sum_{\iota=1}^{10} B[i, n, \iota] \right] \tag{26}$$

The term inside the summation represents an upper bound on the total weight along dimension $i$. Therefore, setting the capacity vector equal to this summation would always allow all items to be selected in the true solution, resulting in a trivial case. To avoid this, we add $r \in [0, 1]$. It tightens the capacity constraints, thereby preventing trivial solutions where all items are feasibly selected. In our experiments, we set $r$ to 0.2. Due to the multiplicative noise term $\xi_i^{(\kappa)} \sim U[1 - w, 1 + w]$, the capacity vectors for each instance are not exactly the same.

#### A.1.2  Data Generation for Unknown Capacity Vectors

We again follow an approach similar to Tang and Khalil [30] to construct this dataset. To generate the capacity vector, we first create a matrix, $B \in \mathbb{R}^{M \times 10}$ is generated, which represents the true underlying model. Each entry of the matrix $B$ is sampled from independent Bernoulli distributions of probability 0.5 like before. The $i$-th dimensional capacity $\rho_i^{(\kappa)}$, for $\kappa$-th instance, is then generated according to the following formula:

$$\rho_i^{(\kappa)} = \left[ \frac{1}{3.5^{\text{Deg}}} \left( \frac{1}{\sqrt{10}} \left( \phi_\kappa^\top B[i] \right) + 3 \right)^{\text{Deg}} + 1 \right] \xi_i^{(\kappa)} \tag{27}$$

Table 1: Test Regret and Infeasibility of the Models on MDKP Weight Prediction.

| Model | Infeasibility | | Regret | |
|---|---|---|---|---|
| | Avg | Sd | Avg | Sd |
| MSE | 0.335 | 0.015 | 0.063 | 0.003 |
| CombOptNet | 0.996 | 0.003 | 0.000 | 0.000 |
| SFL | 0.992 | 0.004 | 0.010 | 0.013 |
| 2sPtO | 0.996 | 0.003 | 0.000 | 0.000 |
| Odece(0.2) | 0.952 | 0.020 | 0.017 | 0.002 |
| Odece(0.3) | 0.845 | 0.019 | 0.024 | 0.002 |
| Odece(0.4) | 0.570 | 0.050 | 0.039 | 0.002 |
| Odece(0.5) | 0.134 | 0.023 | 0.081 | 0.004 |
| Odece(0.6) | 0.079 | 0.027 | 0.110 | 0.009 |
| Odece(0.7) | 0.066 | 0.021 | 0.126 | 0.011 |
| Odece(0.8) | 0.065 | 0.025 | 0.127 | 0.007 |

Table 2: Test Regret and Infeasibility of the Models on MDKP Capacity Prediction.

| Model | Infeasibility | | Regret | |
|---|---|---|---|---|
| | Avg | Sd | Avg | Sd |
| MSE | 0.588 | 0.024 | 0.171 | 0.010 |
| CombOptNet | 0.647 | 0.092 | 0.202 | 0.024 |
| SFL | 0.251 | 0.124 | 0.269 | 0.044 |
| Odece(0.2) | 0.791 | 0.077 | 0.123 | 0.022 |
| Odece(0.3) | 0.572 | 0.269 | 0.175 | 0.086 |
| Odece(0.4) | 0.153 | 0.184 | 0.330 | 0.097 |
| Odece(0.5) | 0.053 | 0.020 | 0.392 | 0.039 |
| Odece(0.6) | 0.049 | 0.024 | 0.412 | 0.058 |
| Odece(0.7) | 0.055 | 0.020 | 0.417 | 0.042 |
| Odece(0.8) | 0.038 | 0.008 | 0.428 | 0.042 |

However, directly using this capacity value would result in capacities that are too low for most instances, leading to solutions where no items are selected. To obviate such scenarios, we scale the capacity vector by a factor $r = 0.5 * N$, where N is the number of items. Further, to preserve the dependency between the feature vector and the capacity vector, we multiply the feature vector by the same scaling factor $r$.

## A.2 Brass alloy production problem

We reproduce this experimental setup from the work by Hu et al. [15]. It is actually a covering LP problem. The production of the Brass alloy requires two metals – Copper (Cu) and Zinc (Zn). A factory needs to purchase ores that contain these two metals. These metals must be sourced from multiple suppliers, each offering metal ores at different prices per unit. The factory's goal is to purchase a combination of ores from these suppliers that minimizes the total cost while ensuring that the total quantity of each metal meets the production requirements. The optimization problem can be formally written as:

$$\min_{x_{1:N}} \sum_{n=1}^{N} q_n x_n \quad \text{such that} \quad x_n \geq 0 \quad \forall n \in [N], \quad \sum_{n=1}^{N} \rho_{ni} x_n \geq \rho_i, \ \forall i \in [M] \tag{28}$$

where the decision variable $x_n$ represents the amount of ore to be purchased from supplier $n$, and $q_n$ is the cost per unit from that supplier. $\rho_{ni}$ denotes amount of metal $i$ in one unit of ore from supplier $n$. The constraints ensure that the total acquired amount of each metal $i$ across all suppliers sums up more than $\rho_i$, the required quantity for production.

Table 3: Test Regret and Infeasibility of the Models on Alloy Production Problem.

| | Infeasibility | | Regret | |
| | Avg | Sd | Avg | Sd |
| Model | | | | |
| --- | --- | --- | --- | --- |
| MSE | 0.532 | 0.005 | 0.169 | 0.004 |
| CombOptNet | 0.896 | 0.233 | 0.181 | 0.002 |
| SFL | 0.991 | 0.013 | 0.019 | 0.001 |
| 2sPtO | 0.400 | 0.548 | 8.539 | 6.691 |
| Odece(0.2) | 0.790 | 0.064 | 0.081 | 0.033 |
| Odece(0.3) | 0.729 | 0.078 | 0.123 | 0.025 |
| Odece(0.4) | 0.571 | 0.085 | 0.195 | 0.041 |
| Odece(0.5) | 0.471 | 0.123 | 0.270 | 0.054 |
| Odece(0.6) | 0.324 | 0.042 | 0.365 | 0.045 |
| Odece(0.7) | 0.276 | 0.026 | 0.406 | 0.037 |
| Odece(0.8) | 0.174 | 0.029 | 0.486 | 0.037 |

In the Brass alloy problem instances, there are 10 customers and the factory needs at least 627.54 units of Cu and 369.72 units of Zn. We consider the data for Brass alloy, which is publicly available in the repository [3]. The dataset contains 500 instances. In each instance, there are 4096-dimensional feature vector for predicting the values of each $\rho_{ni}$. We use 350 instances for training, 50 for validation, and 100 for testing.

# B  Additional Experimental Results

In this section of the Appendix, we provide additional details on the experimental results that could not be included in the main text due to space limitations. Specifically, we report the average and standard deviation of test regret and infeasibility across five runs for each model. Table 1, Table 2 and Table 3 present the average average and standard deviation of test regret and infeasibility across five runs for MDKP weight prediction, MDKP capacity prediction and Brass alloy production problem respectively.

| $\alpha$ | p-value (infeasibility) | p-value (regret) |
| --- | --- | --- |
| 0.8 | $3 \times 10^{-6}$ | 0.999 |
| 0.7 | $3 \times 10^{-5}$ | 0.999 |
| 0.6 | $5 \times 10^{-6}$ | 0.999 |
| 0.5 | $5 \times 10^{-5}$ | 0.999 |
| 0.4 | 0.999 | $1.50 \times 10^{-4}$ |
| 0.3 | 0.999 | $4 \times 10^{-6}$ |
| 0.2 | 0.999 | $3 \times 10^{-6}$ |

Table 4: P-values from paired t-tests comparing MSE and Odece for infeasibility and regret with different values of $\alpha$ for MDKP weight predictions.

**Test of statistical significance.**   For predicting capacity, we observe in Table 5, Odece has significantly lower infeasibility and higher regret till $\alpha = 0.4$. We also observe the same for the alloy production problem in Table 6. We conducted statistical significance tests to determine whether the test regret and infeasibility of Odece are lower than those of MSE. Since each run was conducted using the same random seed for both models, we used a paired t-test with the alternative hypothesis that MSE has higher regret and higher infeasibility than Odece. The p-values from the paired t-tests for different values of $\alpha$ for MDKP weight predictions are reported in Table 4. It tells that for $\alpha \geq 0.5$, Odece has significantly lower infeasibility and higher regret than MSE. The opposite is true when $\alpha$ goes below 0.4.

---

[3]`https://github.com/Elizabethxyhu/NeurIPS_Two_Stage_Predict-Optimize/`

| $\alpha$ | p-value (infeasibility) | p-value (regret) |
|---|---|---|
| 0.8 | $6 \times 10^{-7}$ | 0.995 |
| 0.7 | $2 \times 10^{-7}$ | 0.995 |
| 0.6 | $9 \times 10^{-7}$ | 0.999 |
| 0.5 | $1 \times 10^{-6}$ | 0.997 |
| 0.4 | 0.003 | 0.999 |
| 0.3 | 0.450 | 0.545 |
| 0.2 | 0.997 | 0.040 |

Table 5: P-values from paired t-tests comparing MSE and Odece for infeasibility and regret with different values of $\alpha$ for MDKP capacity predictions.

| $\alpha$ | p-value (infeasibility) | p-value (regret) |
|---|---|---|
| 0.8 | $4 \times 10^{-6}$ | 0.999 |
| 0.7 | $7 \times 10^{-6}$ | 0.999 |
| 0.6 | 0.002 | 0.999 |
| 0.5 | 0.160 | 0.993 |
| 0.4 | 0.848 | 0.882 |
| 0.3 | 0.999 | 0.007 |
| 0.2 | 0.999 | 0.002 |

Table 6: P-values from paired t-tests comparing MSE and Odece for infeasibility and regret with different values of $\alpha$ for Alloy Production Problem.

# C   Details about Hyperparameter Configuration

Finally, in Table 7, we provide the hyperparameter configurations used for each model to reproduce the results reported above.

Table 7: Hyperparameter configurations for each model and task.

| Task | Hyperparameter | MSE | Odece | CombOptNet | SFL | 2sPtO |
|---|---|---|---|---|---|---|
| MDKP Weight Prediction | Learning Rate | 0.05 | 0.05 | 0.05 | 0.05 | 0.05 |
| | $\tau$ (CombOptNet) | | | 0.50 | | |
| | Temperature (SFL) | | | | 0.50 | |
| | damping (2sPtO) | | | | | 0.01 |
| | thr (2sPtO) | | | | | 0.1 |
| MDKP Capacity Prediction | Learning Rate | 0.005 | 0.005 | 0.005 | 0.01 | |
| | $\tau$ (CombOptNet) | | | 0.1 | | |
| | Temperature (SFL) | | | | 0.1 | |
| Brass Alloy | Learning Rate | 0.001 | 0.001 | 0.005 | 0.001 | 0.05 |
| | $\tau$ (CombOptNet) | | | 0.5 | | |
| | Temperature (SFL) | | | | 0.5 | |
| | damping (2sPtO) | | | | | 0.01 |
| | thr (2sPtO) | | | | | 0.1 |

