# OpenReview forum: "Feasibility-Aware Decision-Focused Learning for Predicting Parameters in the Constraints"
_NeurIPS.cc/2025/Conference — NeurIPS 2025 poster_

### Official Review · Reviewer_fGw6 · 2025-07-01

**Clarity:** 2
**Significance:** 2
**Originality:** 3
**Rating:** 4
**Confidence:** 4

**Summary:**

Generally, this paper proposed a trade-off learning approach for decision-focused learning/prediction. The penalization for infeasibility and suboptimality is balanced through a convex combination of them. The effectiveness of the proposed penalization is proven theoretically. Experimental results show the performance of the approach under different trade-off parameters.

The proposed method is well-formulated and empirically supported, and the authors clearly state that "it is important to simultaneously manage both feasibility and decision quality." While the tunable trade-off parameter introduced in the paper provides a flexible way for decision-makers to control this balance, its practical utility could be further improved by offering guidance on how to align the parameter with real-world decision-making preferences. In particular, when constraint parameters are implicit or arise from black-box systems, it becomes nontrivial for decision-makers to calibrate the trade-off between feasibility and suboptimality without a structured mechanism or interpretive support. As acknowledged in the paper, decision-makers may wish to “control the trade-off between suboptimality and infeasibility” according to their preference, and designing data-driven or preference-informed ways to adapt this parameter could enhance the applicability of the approach in practical settings.

**Questions:**

1. The essence of the penalization is the amount of constraint violation and margin, after ReLU approximation. Considering the value range of the different constraint functions is different, a proper normalization approach should be proposed.
2. For equality constraints, the higher buffer (constraint margin parameter) is likely to kill their influence on the loss function. Meanwhile, the decision makers’ concern about the violation direction may be different; this concern could possibly solve this problem.
3. The selection of the convex combination trade-off parameter should be discussed.
4. A little more related work should be discussed and cited.

**Ethical Concerns:**

["NO or VERY MINOR ethics concerns only"]

**Final Justification:**

I would keep my current score, because it is challenging for decision-makers to specify their preferences through the hyperparameter $\alpha$ without clear guidance. Generally, this paper proposed a trade-off learning approach for decision-focused learning/prediction. I'm fine if the paper is accepted.

**Limitations:**

Yes, the biggest limitation of the proposed method is that it requires solving the programming problem during the training process of the prediction model.

**Quality:**

2

**Strengths And Weaknesses:**

The quality of this paper is generally high. The present is clear; perhaps more results should be discussed. The approach proposed in this paper doesn’t show high/strong significance compared with other existing works. From the point of view of the reviewer, the work in this paper is original. Specifically, the derivation of the two loss functions (one penalizes infeasibility and the other penalizes suboptimality) is principled and well-grounded in a maximum likelihood framework. The proposed method is general enough to accommodate nonlinear and integer programs, unlike many prior works restricted to LPs/ILPs. Experiments demonstrate that tuning the trade-off parameter $\alpha$ can achieve a flexible balance between regret and feasibility, which may align with different decision-maker preferences.

While the proposed formulation is sound and the empirical results are convincing, the paper could benefit from a deeper discussion on how decision-makers can select or adapt the trade-off parameter $\alpha$ in practice. This is particularly important because α encodes a key behavioral preference — how to weigh feasibility versus optimality under uncertainty — but is currently treated as an external hyperparameter, selected by trial. More specifically, in practical applications, it is challenging for decision-makers to specify their preferences through the hyperparameter $\alpha$ without clear guidance, which may limit the practical utility of the proposed method.

---

> ### Author Rebuttal · Authors · 2025-07-29
>
> Thank you for your suggestion to expand the discussion on related work.  We will provide a more extensive review of DFL and contextual stochastic optimization by citing the recent survey papers by Mandi et al., JAIR (2024) and Sadana et al., EJOR (2025) and the references therein. The paper closest to ours is the work by Hu et al., who also focus on making point predictions of constraint parameters. However, their approach assumes the existence of a recourse action in case of infeasibility.  We refer to our response to reviewer VPp3, where we highlight how their approach differs from ours. Note that we also show that by increasing $\alpha$, our approach can deliver lower infeasibility than theirs.
>
> - **Normalizing Constraint Violation:** We experimented with normalizing constraint violations $\max\big( \big( \Upsilon + g_i \left( x^\star , \hat{\rho} \right) \big) , 0 \big)$ by dividing by absolute value of the true constraint values $|g_i \left( x^\star, \rho \right)|$. This did not work because in our setting, the absolute magnitude of violation is what directly impacts feasibility. Normalization led to disproportionately smaller penalties for constraints with higher true values. Moreover, constraints with higher values of $|g_i \left( x^\star, \rho \right)|$ are more likely to result in violations with the predicted constraint $\hat{\rho}$. As a result, normalizing by the absolute value of the true constraint reduced the penalty for such constraints, increasing the overall rate of infeasibility.
>
> - **Buffer with equality constraints:** We formulated the problem definition only with inequality constraints. Any equality constraints can be split into two inequality constraints, but in that case the buffer will point in the opposite direction. We have not experimented with equality constraints yet. One possible solution is to keep the buffer zero for equality constraints.
>
> - **Setting $\alpha$:** Our motivation was to leave the selection of the tunable parameter, $\alpha$, on the decision-maker. $\alpha$ controls the trade-off between feasibility and optimality. As mentioned in response to Reviewer pSY2, the choice of parameter can be set empirically, for example based on domain-specific reliability requirements, e.g. tolerance in production lines.

---

> > ### Comment · Reviewer_fGw6 · 2025-08-05
> >
> > Thanks for the response. Most of my concerns are addressed. After reading all reviews and responses, I would like to mantain my current score.

---

> > > ### Author Response · Authors · 2025-08-09
> > >
> > > Thank you for your time, careful consideration, and constructive feedback. We are grateful for your engagement during the discussion period and are glad you found our responses useful.

---

### Official Review · Reviewer_pSY2 · 2025-07-02

**Clarity:** 4
**Significance:** 2
**Originality:** 3
**Rating:** 4
**Confidence:** 4

**Summary:**

The paper is in the domain of Decision Focused Learning, where we have to predict parameters of some optimisation problem and subsequently solve it. Instead of minimising the error in predicted parameters, Decision Focused learning minimises the error in decision outcomes from the optimisation problem. However, in a constrained optimisation problem, if predicted parameters also include the constraints, then such a DFL setup may lead to infeasible solutions. This work proposes DFL in constrained optimisation problems by proposing two loss functions -  Infeasibility Penalty Loss (IPL) and Optimality-Preserving Loss (OPL). By adjusting the weighted combination of these two losses, the end user can decide the tradeoff between feasibility and optimality. The paper also showcases empirical results on two constrained optimisation problems - a multi-dimensional knapsack problem and a covering LP problem. Compared to baselines, the proposed method precisely allows controlling performance in feasibility or optimality dimensions based on the loss combination weight.

**Questions:**

Questions and Comments:

5 seeds seem to be a low number for the empirical experiments. It would be useful to run more seeded experiments.

What would be the kinds of scenarios where a decision-maker would choose feasibility over optimality (or vice versa)? While the paper shows that the proposed technique allows precise control over the tradeoff between the two, it is not motivated why we want this control and further unclear how the decision maker will choose the mixing weight. A decision on these questions would be critical to evaluate the significance of the contributions of this paper.

In the Optimality-Preserving Loss formulation described in lines 214-216, when you replace a feasible point x with x*, it is a bit unclear how much the loss would be sensitive to the choice of the feasible point x?

**Ethical Concerns:**

["NO or VERY MINOR ethics concerns only"]

**Final Justification:**

The authors have described that the dataset and problem setting in the paper have been used to benchmark the Decision Focused Learning technique in multiple prior works as well. They have described the connections between controlling the optimality-feasability tradeoff and Chance-Constrained Optimisation. They have also explained the behaviour of solutions from SFL and given an intuition on why its solutions lead to poorer performance, and particularly why SFL tends to choose a specific style of solutions. Lastly, the authors have pointed to the statistical significance tests they have conducted to show that ODECE performs significantly differently from the baseline methods. These points adequately answer the queries I had, and it would be useful to highlight them in the main paper to strengthen the contributions of this work.

**Limitations:**

yes

**Quality:**

3

**Strengths And Weaknesses:**

Strengths:
The paper is well written and easy to follow, and the problem formulation is comprehensive. Claims on the correctness of proposed loss functions are supported by proofs.

The problem of predicting constraint parameters in optimisation problems as a single-stage decision-focused learning problem is important. Unlike prior work, the proposed method in the paper can handle broad classes of constrained optimisation problems.

Likelihood perspective to DFL, i.e., maximising likelihood of the predicted parameters leading to an optimal solution, is an interesting framing, especially because it encapsulates both the objective function parameters and the constraint parameters.



Weaknesses:

While the paper claims that the method is applicable for a general class of constrained optimisation problems (even outside linear programs and mixed integer programs), the two problem settings in the empirical results seem like somewhat small examples and are rather limited. Prior works in DFL, for instance, have often used real-world data from healthcare, finance, etc,  which model the relationship between real-world features and predicted parameters of optimisation problems. It would be useful to expand the set of experiments to such scenarios.

It would be useful to interpret the reasons why and in what scenarios the SFL or existing method prefer feasibility or optimality. For example, in Figure 1, we see that SFL chooses a solution that balances the tradeoff in two dimensions for the ‘predicting capacities in MDKP’ problem, but it favours minimising regret in the other two problems. No reasoning is provided for why such behaviour happens (and the interpretation of parameters which lead to the proposed method’s control of the tradeoff).

---

> ### Author Rebuttal · Authors · 2025-07-29
>
> Thank you for your review and analysis and the positive assessment that the theoretical claims of our proposed loss functions are well supported by formal proofs.
> - **Regarding the scope of the experimental setup:** We want to clarify that the Brass Alloy Production problem originates from a real use case and is based on real-world data. Moreover, this use case was previously used by Hu et al. in their NeurIPS paper. The Multi-Dimensional Knapsack Problem (MDKP) is adapted from the knapsack experiment in the PyEPO benchmark, which is widely used to evaluate DFL methodologies (e.g., recently by Zharmagambetov et al., NeurIPS (2023); Lin, Delage, and Chan, NeurIPS (2024); Huang and Gupta, NeurIPS (2024)). We used the same underlying relationship between features and optimization parameters as used in these papers, with the only difference being that the predicted parameters appear in the constraints in our work. While more experiments are always of interest, we argue that our current scope is in line with that of existing work and shows the promise of the approach.
>
>
> - **Choosing feasibility over optimality:** We explain in the text that achieving both optimality and feasibility involves a trade-off, as enforcing strict feasibility can lead to overly suboptimal solutions. To manage this, our approach introduces a tunable parameter $\alpha$ that balances the two loss components: IPL (penalizing infeasibility) and OPL (penalizing suboptimality). This formulation allows users to navigate the trade-off according to their preferences. The idea is similar in spirit to chance-constrained optimization (CCO), where no recourse actions are applied and where constraints are instead required to hold up to a specified probability to provide reliability under uncertainty. However, CCO typically assumes that the uncertain parameters follow a known distribution. In contrast, our approach makes point prediction of the constraint parameters without assuming they come from certain distributions. In CCO, the probability level is often chosen based on domain-specific reliability requirements (e.g., acceptable tolerances in production lines). Likewise, $\alpha$ could also be chosen based on domain-specific reliability requirements.
>
> - **Performance of SFL:** We observed that the SFL technique tends to produce low-magnitude gradients for the constraint parameter $a$ (the LHS of constraints of the form $ax \leq b$) and relatively higher gradients value for the RHS parameter $b$. We suspect this behavior is due to the way negative samples are generated. For example, in the MDKP problem, if the optimal solution is [0,0,1] in a 3-dimensional MDKP, the negative samples include [1,1,1],[1,1,0], [0,1,1] and [1,0,1]. These negative samples can be classified as negative based on capacity, the RHS parameter.
> Based on this, we believe that the SFL approach tends to place more importance on the RHS parameter, and the LHS parameter is less frequently used in discriminating between positive and negative samples. This in our view leads to poor performance when SFL is used to estimate only the LHS parameter.
>
>
> - **Experiment with five seeds:** We ran all experiments with five different random seeds and reported both the average and standard deviation in the Appendix. Additionally, we performed statistical significance tests to show that ODECE performs significantly differently from the baseline methods. This provides substantial evidence that our conclusions are valid.
>
> - **Choosing Feasible Point in OPL Loss Formulation:** The performance of the proposed loss indeed depends on the choice of the feasible point used in Eq. (15). We use the true optimal point (with respect to the true parameters) as the feasible point for two main reasons:
>     1.  It is already available in the training set, and hence does not require solving an additional satisfaction problems; this could be expensive depending on the constraints.
>     2. It enables Lemma 2, which guarantees that zero IPL and OPL imply optimality under true parameters—something we couldn’t prove with arbitrary feasible points.

---

> > ### Comment · Reviewer_pSY2 · 2025-08-05
> > **Response**
> >
> > The authors have described that the dataset and problem setting in the paper have been used to benchmark the Decision Focused Learning technique in multiple prior works as well. They have also explained the behaviour of solutions from SFL. It would be useful to include it in the main paper.

---

> > > ### Comment · Reviewer_pSY2 · 2025-08-09
> > >
> > > Wrapping up: The authors have described that the dataset and problem setting in the paper have been used to benchmark the Decision Focused Learning technique in multiple prior works as well. They have described the connections between controlling the optimality-feasability tradeoff and Chance-Constrained Optimisation. They have also explained the behaviour of solutions from SFL and given an intuition on why its solutions lead to poorer performance, and particularly why SFL tends to choose a specific style of solutions. Lastly, the authors have pointed to the statistical significance tests they have conducted to show that ODECE performs significantly differently from the baseline methods. These points adequately answer the queries I had, and it would be useful to highlight them in the main paper to strengthen the contributions of this work.

---

> > > > ### Author Response · Authors · 2025-08-09
> > > >
> > > > Thank you for your time, careful consideration, and constructive feedback. We are grateful for your engagement during the discussion period and are glad you found our responses useful. As NeurIPS allows for one additional page in the final submission, we will incorporate these points into the main text.

---

### Official Review · Reviewer_VPp3 · 2025-07-09

**Clarity:** 2
**Significance:** 3
**Originality:** 3
**Rating:** 4
**Confidence:** 4

**Summary:**

This paper proposes a new Decision-Focused Learning (DFL) approach that specifically addresses learning/predicting constraint parameters for (combinatorial) optimization problems. It should be noted that the vast majority of research on DFL deals with predicting objective function parameters, and only few publications deal with DFL for constraints. Predicting constraint parameters is challenging since in case of hard constraints wrong predictions can lead to infeasible solutions. The paper introduces a new loss function consisting of a convex combination of two (also new) loss functions that explicitly address infeasibility: The first loss function (IAL) aims at minimizing the probability that that the estimated parameters induce a constraint violation, the second loss function (OPL) aims at minimizing the probability that the (true) optimal solution is violated in the optimization problem parameterized by the predicted parameters. The relative weight of the two loss functions in the compound loss function can be adjusted according to the preferences of the decision maker.
In a set of computational experiments with the multidimensional knapsack problem and an alloy production problem, it is shown that by varying the weight parameter of the loss function, it is possible to navigate the trade-off between a small regret and a small probability of yielding infeasible solutions.

**Questions:**

1.	Referring to my comment above, please provide additional results using the approach detailed in Appendix A2 of the Hu et.al. 2023 NeurIPS paper, with varying penalty weights.
2.	Given that also the approach in that paper uses an indicator function for infeasibility, could you please elaborate a bit more on the differences to your approach?
3. How to choose the buffer parameter?

**Ethical Concerns:**

["NO or VERY MINOR ethics concerns only"]

**Final Justification:**

My main concern with the paper was that I found the authors did not perform a thorough comparison against the Hu et al. paper. This issue was addressed in the review, so I am willing to increase my score.

**Limitations:**

Limitations are addressed, with the exception mentioned above that some related work is not represprented and considered adequately.

**Quality:**

3

**Strengths And Weaknesses:**

### Strengths
1.	The paper proposes an innovative approach for DFL with uncertain constraint parameters.
2.	The new loss functions are very adequate and seem to work well
3.	The idea of introducing a parameter for balancing the trade-off between small regrets and probability of infeasible is very good
4.	The computational results show that the approach allows to trade optimality vs feasible using the introduced parameter, and for some settings of the trade-off weight, the approach seems to dominate alternative approaches

### Weaknesses:
1.	For me, an important concern with the paper is that it does not properly account the approach from Hu et al 2023, NeurIPS) (Reference 13 in the submission), both when describing the approach and also in the computational experiment.    While it is true that the approach is designed as a two-stage approach with so-called corrective actions / penalty functions, the paper from Hu clearly describe how to use it in a setting with hard constraints, the details can be found in the Appendix .. of Hu ; they use an indicator function combined with a penalty in order to make predictions inducing infeasibilities less likely. In that appendix, there is even a small set of computational results with the 0/1 knapsack problem in which they show that increasing the penalty increases the empirical probability of achieving feasible solutions (e.g. a probability of >= 99% in case of a penalty of 4.
As a result, I would have expected the authors to include results with the penalty function mentioned in the Appendix of Hu et al., and with much higher penalty values: In the present manuscript, you use small penalty values that were used by Hu et al in a setting with corrective actions, which is not appropriate.
2.	The authors also write that existing approaches do not allow adjusting the prediction based on their subjective preference between optimality and feasibility. Given the previous remark, by adjusting the penalty weight when using the penalty function for hard constraints, the approach does allow that.
3. The "buffer" parameter is introduced in the paper, but the paper gives no hint on how to choose it, although I feel it will have a considerable impact on the prediction.

---

> ### Author Rebuttal · Authors · 2025-07-28
>
> Thank you for your review of our work and clarification questions, which we address below:
>
> **Comparison against Hu et al 2023, NeurIPS:**
> -  We first clarify how our use of an indicator function is very different from the approach by Hu et al:
>     - The approach proposed by Hu et al. consists of two stages. Let the first-stage solution be $x_1^\star$ and the second-stage solution be $x_2^\star$. To apply their approach in a setting where performing a recourse action is not possible, they indeed mention in the appendix, that they impose a high penalty value $P \to \infty$ if $x_2^\star$ differs from $x_1^\star$. So, the indicator function is used to denote whether the *second-stage* solution $x_2^\star$ is different from the first-stage solution be $x_1^\star$, resulting in the loss function: $P * \mathbb{1} [x_1^\star \neq x_2^\star] $. .
>     - On the other hand, we use an indicator function in the IAL loss to denote whether the (first/only stage) assignment is infeasible, i.e., whether it violates any constraints. More specifically, we use an indicator function to denote whether an assignment $x$, violates one specific constraint, e.g., for the $i^\text{th}$ constraint: $g_i \leq 0$, i.e., $\mathbb{1} [g_i(x) >0] $.
>     - Moreover, in the approach of Hu et al., the indicator function $P * \mathbb{1} [x_1^\star \neq x_2^\star] $ is used in the objective function of the second-stage optimization problem. Instead, we only use an indicator function over individual constraints in one of the two losses, after we solve the original optimization problem. So, our approach is very different indeed.
>
> - Secondly, Hu et al. indeed have a short discussion in the appendix on how to apply their method in applications where recourse actions are not possible. As per your suggestion, we ran their approach for different values of penalty $P$ on the Brass Alloy problem, as shown below. We do not observe that the infeasibility ratio monotonically goes down with increasing $P$, as they did for the KP problem. On the other hand, by increasing $\alpha$, we show that our approach can bring down infeasibility rate ($0.14$ at $\alpha=0.8$ in Table 3 in the Appendix), which is lower than any of the values achieved with the method of Hu et al.
>
> | Penalty (P)| Infeasibility  |
> |---------|-------------------|
> | 0.25     | 0.6 $\pm$ 0.55             |
> | 0.5     | 0.76 $\pm$ 0.43               |
> | 4.0     | 0.20 $\pm$ 0.45           |
> | 8.0    | 0.6  $\pm$ 0.55               |
>
> We also started the same experiment on the MDKP problems, but due to the large running time of the Hu et al. method, these have not finished yet.
> - **Regarding the buffer parameter:** We have set the value of the buffer to $\gamma=2$ based on early experiments. It can also be treated as a hyperparameter, where its value would be set based on the result on a validation set, which could even better result, at the computational cost of hyperparameter tuning.

---

> > ### Comment · Reviewer_VPp3 · 2025-08-05
> >
> > Thank you very much for your response, which addresses all my concerns and questions, in particular thank you for your additional experiments which are very helpful.
> >
> > Did your experiments with the MDKP finish at this point?

---

> > > ### Author Response · Authors · 2025-08-08
> > >
> > > Yes, we have run experiments on the MDKP instances as well. Similar to the Alloy dataset, we observe that the infeasibility ratio does not decrease with increasing penalty, as shown below:
> > > | Penalty (P) | Infeasibility  |
> > > |-------------|-----------------------|
> > > | 0.21 (default)        | 0.995                  |
> > > |1       | 0.994                  |
> > > | 2      | 0.994                  |
> > > |4        | 0.994                |
> > > |8        | 0.9930                  |
> > >
> > > For comparison, the infeasibility of our approach ranges from 0.949 ($\alpha=0.2$) to 0.045 ($\alpha=0.8$).
> > >  Since this contradicts the findings of Hu et al., we also tested their method on their 1D knapsack dataset for comparison. We would like to highlight two key differences between their knapsack dataset and ours:
> > > - Dimensionality: Their dataset uses a *1-dimensional knapsack, whereas ours is multi-dimensional.*
> > > - Predicted parameters: They predict *both cost and weight parameters, while in our setting, the objective parameters are fixed, and only constraint parameters are predicted.*
> > >
> > > While running their publicly available code on their 1D knapsack dataset, we noticed that they compute the best result across multiple runs with different initializations, rather than the average across them.
> > > In some runs, feasibility improves with higher penalty. However, if we consider the average across initializations, the results are highly variable -- with feasibility ranging from nearly 0% in one run to 100% in another (even at the same penalty value, e.g., 4). This suggests that their approach is highly sensitive to initialization and suffers from instability. When averaged across runs, we do not observe a consistent improvement in feasibility with increasing penalty even on on their 1D knapsack dataset.
> > >
> > > In contrast, in our experimental evaluation, we compute the average over five independent runs. Our results show that our approach achieves significantly lower infeasibility for $\alpha =0.8$, which other methods — including that of Hu et al. — do not attain.

---

> > > > ### Author Response · Authors · 2025-08-09
> > > >
> > > > Thank you for your time, careful consideration, and constructive feedback. We are grateful for your engagement during the discussion period and are glad you found our responses useful. We appreciate you re-evaluating the paper based on our rebuttal.

---

### Note · Authors · 2025-08-12

Firstly, we sincerely thank all reviewers for their thoughtful comments, suggestions, and constructive feedback.

To summarize our contribution, we have proposed a novel decision-focused learning (DFL) methodology, Odece, for predicting parameters that appear in the constraints of optimization problems. We experimentally demonstrate that by tuning a single parameter, $\alpha$, a decision-maker can achieve very low infeasibility ratios -- levels that none of the competing methods can match.

We are pleased that the reviewers recognized the **novelty of our contribution, the clarity of presentation, and the theoretical soundness of our approach with proofs.**
- Reviewer VPp3 suggested a comparison against Hu et al. (2023). In our rebuttal, we clarified how our proposed approach differs substantially from theirs. We  conducted additional experiments by running the approach of Hu et al. with different penalty values $(P)$ on both the Brass Alloy and MDKP problems. In these experiments, we did not observe their approach significantly reducing the infeasibility ratio with increasing penalty $P$. In contrast, in our approach, increasing $\alpha$ allows us to reach very low infeasibility ratios — a result not attainable with other methods. We hope Reviewer VPp3 will take these findings into consideration when re-evaluating our paper.
- Reviewers fGw6 and pSY2 ask on the feasibility–optimality trade-off and the choice of $\alpha$ in practice. As noted in our rebuttal, achieving both requires balancing the two losses, which motivates introducing $\alpha$ to balance the two losses. This allows decision-makers to tune $\alpha$ to meet domain-specific needs.
- Reviewer pSY2 also ask regarding the performance of SFL. We agree to add our explanation in the main text to help readers understand how Odece differs from SFL.
- We will provide a more extensive review of DFL and contextual stochastic optimization, as Reviewers fGw6 suggested to broaden the discussion on related work.

All other questions not explicitly mentioned here have been addressed in our rebuttal. We appreciate the opportunity to respond and thank all the reviewers for their careful consideration and engagement.

---

### Decision · Program_Chairs · 2025-09-17

**Decision:**

Accept (poster)

**Comment:**

This paper presents a decision-focused learning (DFL) framework for predicting constraint parameters in constrained optimization problems (in contrast to the majority of DFL research, which deals with predicting objective function parameters). As constraint parameter prediction may lead to issues regarding optimization feasibility, the submission introduces a mixed loss function penalizing both infeasibility and solution optimality, with a weighting between them that can be chosen by a decision-maker. Experiments on multi-dimensional knapsack and brass alloy production problems show compelling performance compared to baselines, as well as the ability to indeed trade off between feasibility and optimality via choice of the weighting parameter.

Strengths: Reviewers felt the proposed approach is innovative -- e.g., one reviewer noted that the likelihood perspective to DFL is an interesting framing, and a number commended the conceptual focus on constraints and on broad classes of problems. The computational results are informative in demonstrating the efficacy of the approach in terms of overall performance and trading off between feasibility and optimality. The writing and presentation are clear.

Weaknesses: Reviewers felt it to be a strong assumption that the decision-maker would know the precise desired tradeoff between feasibility and optimality -- this merits additional discussion in the submission. They also felt the number of experimental settings was more limited than it could have been, and had questions regarding the extent to which this framework is more broadly applicable in practice.

Other major discussion points:
* Reviewer VPp3 initially pointed out the lack of comparison against Hu et al (2023) - however, this was resolved in the reviewer-author discussion via clarifications and an additional experiment by the authors. The authors should include these additional experiments (for both multi-dimensional knapsack and brass alloy) in the revised paper.
* Reviewer pSY2 initially pointed out a lack of explanation regarding when/why the SFL baseline method chooses to prioritize feasibility vs. optimality in different settings, as well as the connections between controlling the optimality-feasibility tradeoff and chance-constrained optimization. The authors addressed this in the discussion, and indicated they would update the paper to address these points.
* Reviewer pSY2 initially also pointed out that the authors run experiments over 5 seeds, but the authors clarified that they have run statistical significance tests to show their method's improvement is indeed significant.

Overall, this seems to be an interesting and technically solid contribution, and while there is room for improvement, the strengths are sufficient to warrant recommendation of acceptance.